# *Arabidopsis thaliana* G3BP Ortholog Rescues Mammalian Stress Granule Phenotype across Kingdoms

**DOI:** 10.3390/ijms22126287

**Published:** 2021-06-11

**Authors:** Hendrik Reuper, Benjamin Götte, Lucy Williams, Timothy J. C. Tan, Gerald M. McInerney, Marc D. Panas, Björn Krenz

**Affiliations:** 1Plant Virus Department, Leibniz-Institut DSMZ-Deutsche Sammlung von Mikroorganismen und Zellkulturen GmbH, 38124 Brunswick, Germany; hendrik.reuper@dsmz.de; 2Department of Microbiology, Tumor and Cell Biology, Karolinska Institutet, 171 77 Stockholm, Sweden; benjamin.gotte@yale.edu (B.G.); lucy.williams99@outlook.com (L.W.); TANT0113@e.ntu.edu.sg (T.J.C.T.); gerald.mcinerney@ki.se (G.M.M.); marc.panas@ki.se (M.D.P.)

**Keywords:** stress granules, G3BP, *Arabidopsis thaliana*

## Abstract

Stress granules (SGs) are dynamic RNA–protein complexes localized in the cytoplasm that rapidly form under stress conditions and disperse when normal conditions are restored. The formation of SGs depends on the Ras-GAP SH3 domain-binding protein (G3BP). Formations, interactions and functions of plant and human SGs are strikingly similar, suggesting a conserved mechanism. However, functional analyses of plant G3BPs are missing. Thus, members of the *Arabidopsis thaliana* G3BP (AtG3BP) protein family were investigated in a complementation assay in a human G3BP knock-out cell line. It was shown that two out of seven AtG3BPs were able to complement the function of their human homolog. GFP-AtG3BP fusion proteins co-localized with human SG marker proteins Caprin-1 and eIF4G1 and restored SG formation in G3BP double KO cells. Interaction between AtG3BP-1 and -7 and known human G3BP interaction partners such as Caprin-1 and USP10 was also demonstrated by co-immunoprecipitation. In addition, an RG/RGG domain exchange from Arabidopsis G3BP into the human G3BP background showed the ability for complementation. In summary, our results support a conserved mechanism of SG function over the kingdoms, which will help to further elucidate the biological function of the Arabidopsis G3BP protein family.

## 1. Introduction

Stress granules (SGs) are cytoplasmic RNA/protein structures that assemble in response to environmental stress and contribute to the rapid change of translation from housekeeping genes to stress-response genes. They are membraneless organelles and highly dynamic in terms of their formation in response to various external stresses and subsequent polysome disassembly as well as their disassembly when ambient conditions are restored. In this way, largely preassembled translation complexes can be rapidly released to resume gene expression when cellular stress conditions abate. The scenario most often described for SGs formation follows when oxidative, nutrient or heat stress or viral infection activates one of the eukaryotic initiation factor 2α (eIF2α) kinases, which phosphorylate the alpha subunit of eIF2, leading to reduced levels of eIF2-GTP-tRNAi ternary complexes and the subsequent accumulation of abortive translation initiation complexes on mRNA molecules. Recent studies have proposed a network model of SG assembly, highlighting the importance of a large set of interactions between different proteins and mRNAs as well as liquid–liquid phase separation (reviewed in [1]). A critical regulator of SG assembly is Ras-GAP SH3 domain-binding protein (G3BP; [2,3,4]), a multifunctional RNA-binding protein that is present in two forms in humans, G3BP-1 and -2 [5]. Only when both G3BPs were knocked out was an impairment of SG formation observed in mammalian cells, highlighting G3BP’s crucial role [3,6].

Emerging evidence indicates that plant cells utilize SGs for posttranscriptional gene control, similar to mammalian cells. SG-like structures in plants were identified by cellular-localization studies of eukaryotic initiation factor 4E (eIF4E), oligouridylate-binding protein 1 (UBP1), poly-A binding protein (PABP) and small ribosome subunit proteins [7,8]. In Arabidopsis, a tandem zinc finger protein, AtTZF1, shuttles into SGs-like structures [8]. SG formation in plants has been reported as a stress response to different conditions such as heat, hypoxia or high salt concentrations (reviewed in [9]). It has also been shown that, similar to mammalian viruses, e.g., Semliki Forest virus (SFV), chikungunya virus (CHIKV) and poliovirus [10,11,12], plant viruses can inhibit SG formation [13]. These results indicate that plant cells can assemble SGs that are similar in composition, function and assembly mechanism to those in mammalian cells, but much remains to be investigated.

Human G3BPs (HsG3BP) are multidomain proteins containing a nuclear transport factor 2 (NTF2)-like domain at the N-terminus, an acidic and proline-rich region in the central part of the protein, an RNA recognition motif (RRM) and arginine and glycine-rich region (RG/RGG domain) at the C-terminal end [2]. It was shown that SG formation requires the NTF2-like and RG/RGG domains [2], and recently it has been demonstrated that these domains are also required for CHIKV replication [14,15]. Sequence analyses revealed seven G3BP-like proteins in *Arabidopsis thaliana* [16,17], named nuclear transport factor 2 and RNA recognition motif domain-containing proteins (NTF2-RRM proteins). One of the *Arabidopsis* homologs of human G3BP-1 (named AtG3BP-7 in this study) was shown to be a key regulator of stomatal and apoplastic immunity [16]. Expression of the *A. thaliana* G3BP-like proteins (AtG3BPs) in fusion with fluorescence markers showed recruitment of all individual AtG3BPs to cytoplasmic SGs after heat stress, as well as co-localization with ubiquitin carboxyl-terminal hydrolase 24 (AtUBP-24) in bimolecular fluorescence complementation (BiFC) experiments *in planta* [17,18]. Furthermore, AtG3BP-2 co-localized with the characterized plant SG marker protein TZF1 [18]. AtUBP-24 is a putative homolog of the human ubiquitin-specific peptidase 10 (HsUSP10). HsUSP10 is known to interact with HsG3BP in the mammalian system [19]. HsUSP10 harbors an FGDF motif at its N-terminus and it was shown that this motif is necessary to bind to HsG3BP [20]. The D12A mutations in FGDF motifs are tolerated for binding to HsG3BP, and AtUBP-24 harbors a similar motif (FGSF) at its N-terminus [18]. Consequently, some or all of the seven AtG3BPs may play a functional role in plant SGs as their interaction with AtUBP-24 has been shown.

Due to the lack of an *Arabidopsis* plant line in which all seven AtG3BP protein family members have been successfully knocked out, it is to date not possible to study a SG phenotype in plants, as in the mammalian system. The aim of this study was, therefore, to evaluate the potential of the different members of the AtG3BP family to rescue SG formation in human cells lacking G3BP-1 and -2 (HsG3BP1/2^−/−^ U2OS cells, [6]). AtG3BPs were stably overexpressed in this cell line and stress treatments were performed to investigate which of the AtG3BPs can rescue the knock-out phenotype. We show that AtG3BP-1 and -7 were able to rescue SG formation in human cells. Further, the AtG3BP-complemented human cell lines were investigated for human interaction partners Caprin-1 and USP10 [19,20,21]. Moreover, the C-terminal RG/RGG domain of HsG3BP1 was exchanged with one of the seven C-terminal AtG3BP RG/RGGs to investigate how the conserved domain affects the ability of HsG3BP-1 to form SGs. Our results show that all C-termini that actually contain an RG/RGG domain can form SGs in combination with the N-terminal part of HsG3BP-1. In summary, we propose that the cellular processes underlying the formation of SGs during (a)biotic stress are conserved between mammals and plants.

## 2. Results

### 2.1. Phylogenetic Relationship of HsG3BPs and the Arabidopsis thaliana G3BP Family

Protein sequence alignment of the human G3BP family, consisting of HsG3BP-1, HsG3BP-2a and its splicing variant HsG3BP-2b, and the seven members of the *Arabidopsis thaliana* G3BP family, was performed (Figure 1A). The alignment shows that all AtG3BPs and HsG3BPs share a conserved N-terminal NTF2-like domain and a C-terminal RRM domain. Additionally, several amino acids, e.g., phenylalanine at positions 23 and 41 of the consensus sequence, which refer to positions 15 and 33, respectively, in the HsG3BPs, are conserved in all G3BPs, but F41 (consensus) is exchanged to a Leucine (L) in AtG3BP-4 (Appendix A). These amino acids have been shown to be part of the hydrophobic FGDF-motif binding pocket [22] and to be important for mammalian interaction partners USP10 and Caprin-1 as well as SFV and CHIKV non-structural protein 3 (nsP3) [6,14,20]. Furthermore, a C-terminal RG/RGG domain, containing one or several RG/RGG motifs, is conserved, again except for AtG3BP-4. The acidic and proline-rich region, which is shared by all HsG3BPs, could not be annotated in the AtG3BP family. The highest protein sequence similarity between a member of the human and the Arabidopsis G3BP family is shared by HsG3BP-2b and AtG3BP-6 with 48.2% (Appendix A). At the nucleotide level, HsG3BP-2a and AtG3BP-7 have the highest percentage of identity at 38.4%. To gain better insights into the relationship between the two families, a phylogenetic analysis using the neighbor-joining method was conducted [23]. In all of the bootstrap replicates, the two families are separated into two distinct clusters: the Arabidopsis family and the human family (Figure 1B). These clusters do not change when only the NTF2-like (Appendix A) or the RRM (Appendix A) domains are taken into consideration. For the RG/RGG domain only, AtG3BP-1 and -7 cluster together with the human G3BP family, whereas the remaining AtG3BPs, except AtG3BP-4, which does not harbor an RG/RGG domain, still form a branch (Figure 1C).

### 2.2. Expression of EGFP::AtG3BP in HsG3BP1/2^−/−^ U2OS Cells

Considering the high level of domain structure homology between the two families and the likely conserved functions and dynamics of SG assembly, the question arose of possible complementation of HsG3BP by AtG3BPs in HsG3BP1/2^−/−^ U2OS cells. To address this, the different AtG3BPs were cloned in-frame with EGFP into a mammalian expression vector (pEGFP-C1). After verification of expression in HsG3BP1/2^−/−^ U2OS cells by transient transfection, stably transformed cell lines, each expressing one EGFP::AtG3BP construct, were generated [6], treated with 200 µM sodium arsenite (SA) to induce SG formation and then fixed and stained for the SG markers Caprin-1 and eIF4G1. Fluorescence microscopy revealed SG formation in cells expressing EGFP::HsG3BP-1, as expected and those expressing EGFP::AtG3BP-1 and -7. The GFP signal in these cells co-localized with fluorescence signals for both Caprin-1 and eIF4G1. The other cell lines (expressing EGFP::AtG3BP-2 through -6) showed diffuse GFP signal and no granule-like structures were observed after SA treatment (Figure 2B). Quantitative analysis of 400 cells per construct and condition (100 cells per construct/condition in four independent experiments) revealed SG rescue under stress conditions in 87% of the cells expressing EGFP::HsG3BP-1, 29% in EGFP::AtG3BP-1 expressing cells and 49% of the cells transfected with EGFP::AtG3BP-7 (Figure 2C).

### 2.3. Interaction Studies of the AtG3BPs with Human Cellular Factors

HsG3BP-1 interacts with several proteins, including HsUSP10, HsCaprin-1 and the small ribosomal subunit protein HsRPS6 [6,19,21]. To gain further insight into the complementation capabilities of the various AtG3BPs, these interaction partners were analyzed. Therefore, HsG3BP1/2^−/−^ U2OS cells were transiently transfected with the EGFP::AtG3BP constructs and co-immunoprecipitation was performed to detect interaction partners HsUSP10, HsCaprin-1 and HsRPS6. The results showed that AtG3BP-1, -2 and -7 interacted with HsUSP10 (Figure 3), but to a much lower extent than HsG3BP-1. HsUSP10 copurified with AtG3BP-2 migrated faster in the gel. HsCaprin-1 weakly co-immunoprecipitated together with AtG3BP-1 and -7. Again, this interaction was also considerably weaker than the interaction of HsCaprin-1 with HsG3BP-1 (Figure 3 and Appendix A). HsRPS6 co-immunoprecipitated with AtG3BP-1, -4, -5 and weakly with -7. This assay has also been carried out in the stably transformed cell lines. In three independent experiments, the co-immunoprecipitation of HsCaprin-1 by AtG3BP-1 and -7 could be observed, whereas the interaction of the other AtG3BPs with HsCaprin-1 appeared to be inconsistent (Appendix A, respectively).

### 2.4. Exchange of RG/RGG Domains and SG Induction

Comparison of the complementation experiments (Figure 2) and the phylogenetic analysis (Figure 1) showed that only AtG3BP-1 and -7, which both cluster into one group with the human G3BPs in the phylogenetic analysis of the RG/RGG domains, can rescue SGs. Recently, the RG/RGG domain, together with the NTF2-like domain of HsG3BP-1, has been shown to be necessary for the formation of a complex between HsG3BP-1, the 40S ribosomal subunit and the nsP3 protein of chikungunya virus (CHIKV) [14]. This complex is essential for the replication of CHIKV in human cells. G3BP1 constructs lacking the RG/RGG domain do not interact with the 40S ribosomal subunit and consequently do not support SG formation. These findings, combined with our complementation experiments (Figure 2), led us to substitute the RG/RGG domain of HsG3BP-1 with those of the AtG3BPs to get more insight into the roles of this domain in SG formation. Therefore, the RG/RGG domains of the different AtG3BPs, as well as the corresponding C-terminal end of AtG3BP-4 (called AtG3BP-4-Cterm hereafter), were attached to the C-terminal end of an EGFP::HsG3BP-1 RGG deletion mutant (the corresponding amino acid regions are listed in Appendix A). The resulting fusion proteins, as well as wild-type HsG3BP-1 and HsG3BP-1-∆RGG deletion mutant, were transiently expressed in HsG3BP1/2^−/−^ U2OS cells followed by SA treatment (Figure 4). We observed that all proteins containing RG/RGG domains were able to rescue SG formation. Only HsG3BP-1 ∆RGG and HsG3BP-1-AtG3BP-4-Cterm showed diffuse, cytoplasmic GFP signals after SA treatment. The cells were also stained against the human SG marker protein T-cell-restricted intracellular antigen-1 (TIA-1), which co-localized with the granular GFP signals, verifying the rescue of bona fide SG.

## 3. Discussion

SGs are ribonucleoprotein condensates that form in response to translational arrest induced by biotic or abiotic stress [28]. There is increasing evidence that plant cells also use SGs for posttranscriptional gene control in a similar way to mammalian cells, because SGs are evolutionarily conserved and have been reported in animal [29], plant [30] and yeast cells [31]. The proteomic composition of SGs in yeast and humans [32] is comparable to each other and this is also true for the proteomic analysis for *A. thaliana* [33]. While some proteins have been shown to be part of the SG formation cascade in mammalian cells, such as TIA-1, TIA-1-related (TIAR) proteins [34] and Tudor staphylococcal nucleases (TSNs; [35]), only a double knock-out of HsG3BP1- and -2 leads to a complete lack of SG assembly, highlighting its central role [3]. Recent studies have revealed the roles of related proteins in plants; for example, RBP45/47 and UBP1 are most closely related to the animal TIA-1/TIARs. Of the Arabidopsis RBP45/47 family, RBP47B relocates from the nucleus to cytoplasmic foci in response to heat, salt and hypoxic stress [7,36,37,38]. UBP1B localizes to SGs under heat stress and plays a crucial role in phytohormone signaling and heat-stress tolerance [39,40]. AtG3BP-2 condenses into SGs and interacts with the nuclear shuttle protein of the begomovirus Abutilon mosaic virus and Pea necrotic yellow dwarf virus upon heat stress [18]. However, the roles of plant G3BP-like proteins in SG nucleation and mRNA stabilization await further studies. The G3BPs in *A. thaliana* have not been sufficiently studied, although the assembly and proper function of SGs are dependent on the key enzyme G3BP in mammals.

Human cells lacking G3BP cannot form SGs in response to eIF2α phosphorylation or eIF4A inhibition [6]. Similar data is missing for plants, because simultaneous genetic ablation of all seven G3BPs in *A. thaliana* has not yet been generated. Abulfaraj and colleagues [16] generated and analyzed AtG3BP-7 (AT5G48650) overexpression cell lines and KO lines which showed no phenotype compared to control plants. Our preliminary observations suggest this is also true for the other members of the AtG3BP gene family (data will be presented elsewhere). This might be due to redundancy in the function of AtG3BP whereby one or several AtG3BP proteins could compensate for the loss of one of the others. This is the case in human cells, whereby HsG3BP-1 can compensate for the loss of HsG3BP-2 and vice versa [6]. However, the question remains if the plant homolog(s) of HsG3BP functions in a similar way.

The aim of this study was, therefore, to analyze whether the function of AtG3BPs can be deduced from the mammalian system and, consequently, whether G3BP homologs from *Arabidopsis thaliana* can rescue SG phenotype in HsG3BP1/2^−/−^ U2OS cells. A phylogenetic analysis was performed [16] and the degree of sequence homology suggested functional similarity between the human and *A. thaliana* G3BPs. Indeed, two of seven AtG3BPs were able to restore SG formation, namely, AtG3BP-1 and -7. Interestingly, AtG3BP-2 is based on protein-sequence similarity, more closely related to HsG3BP-1, but could not restore SG formation (Appendix A). AtG3BP-1, -2 and -7 bind to HsUSP10, AtG3BP-1, and -7 show reproducible interaction with HsCaprin-1. AtG3BP-1, -4, -5 and, to a lower extent, -7, interact with HsRPS6. Interestingly, AtG3BP-2 interacts with a faster-migrating HsUSP10 protein and, based on previous findings that full-length HsUSP10 and peptides containing an FGDF motif can block SG formation; this may explain why AtG3BP-2 is unable to restore SGs. The reason for the faster migration of the HsUSP10 band in the AtG3BP-2 co-IP needs to be further investigated. Clearly, all AtG3BPs interact to a lesser extent with either HsUSP10 or HsCaprin-1 compared to HsG3BP, which is most likely unrelated to the overall abundance of the different G3BPs. The interaction of AtG3BP-1, -4, -5 and -7 with RPS6 is interesting because it has been shown that the HsG3BP-1 and HsRPS6 interaction is dependent on the HsG3BP-1 RG/RGG domain [14]. AtG3BP-4 shows a clear interaction with HsRPS6, despite not containing an RG/RGG domain, whereas AtG3BP-2, -3 and -6 contain this domain but do not interact with HsRPS6. Taken together, the only AtG3BP proteins that show interaction patterns broadly similar to HsG3BP-1 with all tested interaction partners are AtG3BP-1 and -7. Especially the interaction with Caprin-1 is noteworthy, as Caprin-1 binding promotes SG formation, which might explain why these two plant proteins can rescue SG formation in human cells.

Besides these protein–protein interactions, another key feature of SG formation is the interaction between G3BPs and RNA through the RRM and RG/RGG domain, with the latter being required for G3BP-mediated SG formation [6]. This is not yet fully understood, but recent research helped to shed some light on this subject. It has been shown that the RG/RGG motif of HsG3BP-1 is an essential factor for host-translation machinery recruitment by different alphaviruses [14] and an HsG3BP-1-∆RGG deletion mutant failed to rescue SG formation when expressed in HsG3BP1/2^−/−^ U2OS cells [6]. Furthermore, Guillén-Boixet and colleagues [4] demonstrated that under non-stress conditions, the HsG3BP-1 RG/RGG motif forms an intramolecular interaction with the intrinsically disordered acidic tract. In stressed cells, unfolded mRNA outcompetes this auto-inhibitory interaction, and SGs are formed. Differences in the interaction of the AtG3BP RG/RGG domains with the acidic domain of HsG3BP-1 might be a possible explanation why AtG3BP-2, -3, -5 and -6 do not rescue SG formation when expressed in full-length but show SGs in the RG/RGG domain swaps. All G3BPs analyzed in this study, except AtG3BP-4, contain an intrinsically disordered region (IDR), the RG/RGG motifs. IDRs are key elements of RNA binding and viral replication (reviewed in [41]). By swapping the RG/RGG domains to the HsG3BP backbone, we could show that the six AtG3BP RG/RGG domains are interchangeable with the human RG/RGG domain, while the C-terminal end of AtG3BP-4 is not. The RG/RGG domains can therefore be excluded as the sole determinant of the SG rescue phenotype shown in Figure 2. The RG/RGG domain substitution construct containing the C-terminal region of AtG3BP-4 showed the same phenotype as HsG3BP-1 ∆RGG due to a lack of RG/RGG motifs, highlighting the importance of those motifs. It should be emphasized that AtG3BP-4, which does not contain an RG/RGG domain and could not form SGs in any of the experiments conducted here, can nevertheless form SGs in the plant system [17].

In summary, these experiments have shown that there is some functional redundancy in the mechanism of SG formation by the G3BP proteins in plants and mammals. Expertise, results and conclusions gained from experiments in the mammalian system can thus be translated onto plants. Therefore, it must be elucidated in the future if the loss or overexpression of one or more of the AtG3BPs can mediate better stress tolerance or drives the plant to become more stress-sensitive compared to a wild-type plant. Then, it will be important to transfer the gained knowledge from model plants, e.g., *Arabidopsis thaliana* or *Nicotiana benthamiana*, to agro-economically important crops, for example, tomato and cassava.

## 4. Materials and Methods

### 4.1. Cell Culture

The permanent cell line U2OS (human osteosarcoma cells, ATCC HTB-96) were maintained at 37 °C and 5.0% CO_2_ in high glucose DMEM (Sigma-Aldrich, Darmstadt, Germany) containing 10% FBS (Sigma-Aldrich, Darmstadt, Germany), 100 U/mL penicillin (Sigma-Aldrich, Darmstadt, Germany) and 100 µg/mL streptomycin (Sigma-Aldrich, Darmstadt, Germany). Stable U2OS-derived double KO ∆∆G3BP1/2 cells constitutively expressing GFP tagged HsG3BP1-WT were obtained from Nancy Kedersha [6]. The cell lines stably expressing AtG3BP1-7 were generated by transfection of HsG3BP1/2^−/−^ U2OS cells, followed by selection in Geneticin (500 µg/mL, Invitrogen, Carlsbad, CA, USA). Stress granule formation was induced with sodium arsenite (Sigma-Aldrich, S7400, Darmstadt, Germany) treatment (200 µM for 60 min or 500 µM for 30 min) before fixation.

### 4.2. Immunofluorescence and Microscopy

Cells were grown on 12 mm glass coverslips (VWR International, Radnor, PA, USA) and fixed using 3.7% formaldehyde (Sigma-Aldrich, 47608, Darmstadt, Germany) (*v/v*) in PBS for 10 min followed by immersion in ice-cold methanol for 10 min, washing with PBS and blocking with 5% horse serum diluted in PBS for 1 h at RT. Primary antibodies were diluted in a blocking buffer as listed below and incubated for 1 h at RT. Glass coverslips were then washed and incubated in a blocking buffer containing secondary antibodies and Hoechst 33,258 (1 µg/mL, Invitrogen, Carlsbad, CA, USA) for 1 h at RT. Washed coverslips were mounted on microscope slides using a vinol mounting medium and analyzed using a Zeiss Axiovert 200M 63x. Images were processed and compiled using Photoshop (Adobe, San José, CA, USA). Cells were scored for SGs by manual counting using fluorescent microscopy. Only cells with granules co-staining for eIF4G1 and Caprin1 with a minimum of three granules per cell were required to score positively. Statistical differences between the two groups in immunofluorescence experiments were evaluated using a nonparametric Mann–Whitney *U* test. *p* < 0.05 was considered significant. Statistical analyses were performed using GraphPad Prism (GraphPad Software, San Diego, CA, USA). Primary antibodies: mouse anti-eIF4G1 (Santa Cruz, sc-133155; 1:400, Dallas, TX, USA), goat anti-TIA-1 (Santa Cruz, sc-1751, 1:100) and rabbit anti-Caprin-1 (Proteintech Group, 15112-1-AP, 1:200, Manchester, United Kingdom). Secondary antibodies: donkey anti-mouse labeled with Alexa Fluor 568 (Invitrogen A10037, 1:1000), donkey anti-rabbit labeled with Alexa Fluor 568 (Invitrogen A10042, 1:1000), donkey anti-goat labeled with Alexa Fluor 568 (Invitrogen A11057, 1:1000 and donkey anti-goat labeled with Alexa Fluor 647 (Invitrogen A21447: 1:200).

### 4.3. Plasmids and Transfection

The coding regions of AtG3BP-1 (AT5G60980), AtG3BP-2 (AT5G43960), AtG3BP-3 (AT3G25150), AtG3BP-4 (AT1G69250), AtG3BP-5 (AT1G13730), AtG3BP-6 (AT2G03640) and AtG3BP-7 (AT5G48650) were amplified by PCR using the primers in Appendix A. The open reading frames (ORFs) of these genes were ligated into pEGFP-C1 (Clonetech, Mountain View, CA, USA) via the *Eco*RI-*Bam*HI sites (AtG3BP-1, AtG3BP-5, AtG3BP-6 and AtG3BP-7) or the *Xho*I-*Bam*HI sites (AtG3BP-2, AtG3BP-3, AtG3BP-4). Plasmids for the RG/RGG domain exchange experiments were generated by amplifying the backbone of pMC-gtGTU2 (FIT Biotech Plc, Tampere, Finland) [42,43] EGFP::HsG3BP-1 ∆RGG and the selected regions of the different Arabidopsis G3BPs via PCR with the respective overhangs (primers are listed in Appendix A). The PCR fragments were then assembled with NEBuilder^®^ HiFi DNA Assembly Master Mix according to the manufacturer’s protocol. All plasmids were verified by sequencing (Eurofins Genomics, Ebersberg, Germany). Cells were transfected with Lipofectamine 3000 (Thermo Fisher Scientific, Waltham, MA, USA) according to the manufacturer’s protocol.

### 4.4. Immunoprecipitation and Western Blot

#### 4.4.1. Transient Transfection

HsG3BP1/2^−/−^ U2OS were grown in 60 mm dishes to a confluency of 80–90% and transiently transfected with EGFP-tagged AtG3BP1-7 or HsG3BP-1 using Lipofectamine 2000 according to the manufacturer’s protocol. After 24 h, transfected cells were washed with cold PBS and scrape-harvested at 4 °C with 400 µL EE-buffer (50 mM HEPES, pH 7.0, 150 mM NaCl, 0.5% NP-40, 10% glycerol, 5 mM EDTA, 2.5 mM EGTA, 0.1 mg/mL Heparin (H3149, Sigma), 1 mM DTT, HALT protease inhibitor). Harvested cells were rotated for 10 min at 4 °C and sonicated for 8 min in an ice-water bath and cleared by centrifugation for 10 min at 10,000× *g* at 4 °C. The EGFP-tagged fusion proteins were then immunoprecipitated with 20 µL GFP-Trap agarose (Chromotek, Planegg-Martinsried, Germany) at 4 °C for 60 min under constant rotation. Beads were washed 4 times in EE lysis buffer and eluted directly into 2x NuPAGE LDS sample buffer and denatured for 5 min at 95 °C. Proteins were resolved in 4–12% Bis-Tris NuPAGE gels (Thermo Fisher Scientific, Waltham, CA, USA) and transferred to nitrocellulose membranes using the Trans-Blot Turbo Transfer system (Bio-Rad, Hercules, CA, USA). Nitrocellulose membranes were blocked using 5% skimmed milk powder in TBST (0.05% Tween 20) and incubated with primary antibody (as listed below) for 1 h at 4 °C followed by incubation with HRP-linked secondary antibody (1 h at RT) in 1% bovine serum albumin in TBST. Chemiluminescence was detected using SuperSignal West Pico substrate (Thermo Fisher Scientific, Waltham, CA, USA) and a ChemiDoc XRS+ Imaging system (Bio-Rad, Hercules, CA, USA). All original blot images can be found in the supplements.

#### 4.4.2. Stable Expression

HsG3BP1/2^−/−^ U2OS stably expressing EGFP-tagged AtG3BP-1 to -7 or HsG3BP-1 were grown to 80–90% confluency on 100 mm dishes, washed with cold PBS and scrape-harvested at 4 °C with 600 µL EE-buffer (50 mM HEPES, pH 7.0, 150 mM NaCl, 0.5% NP-40, 10% glycerol, 5 mM EDTA, 2.5 mM EGTA, 0.1 mg/mL Heparin (H3149, Sigma-Aldrich, Darmstadt, Germany), 1 mM DTT, HALT protease inhibitor). Harvested cells were rotated for 10 min at 4 °C and sonicated for 8 min in an ice-water bath and cleared by centrifugation for 10 min at 10,000× *g* at 4 °C. The EGFP-tagged fusion proteins were then immunoprecipitated with 30 µL anti-GFP sepharose beads [44] at 4 °C for 60 min under constant rotation. The beads were washed 3 times in a cold EE-buffer followed by elution in a 40 µL SDS-sample buffer. Proteins were resolved in a NuPAGE 4–12% Bis-Tris polyacrylamide gel (Invitrogen, Carlsbad, CA, USA) and transferred to an Amersham Hybond P 0.45 PVDF membrane (GE Healthcare, Chicago, IL, USA). PVDF membranes were blocked using 5% skimmed milk powder in TBST (0.05% Tween 20) and incubated with the primary antibody (as listed below) for 16 h at 4 °C followed by incubation with the HRP-linked secondary antibody (1 h at RT) in 1% bovine serum albumin (BSA) in TBST. Chemiluminescence was detected using SuperSignal West Pico substrate (Thermo Fisher Scientific, Waltham, CA, USA) and a ChemiDoc XRS+ Imaging system (Bio-Rad, Hercules, CA, USA). All original blot images can be found in the supplements.

Primary antibodies: rabbit anti-GFP (Abcam, ab290; 1:3,000, Cambridge, United Kingdom), rabbit anti-USP10 (Bethyl, A300-900A; 1:500, Montgomery, TX, USA), mouse anti-RPS6 (Santa Cruz, sc-74459, Dallas, CA, USA), mouse anti-GAPDH (Santa Cruz, sc-166545, 1:1000, Dallas, TX, USA) and rabbit anti-Caprin1 (Proteintech Group, 15112-1-AP; 1:2000, Manchester, United Kingdom). Secondary antibodies: HRP-conjugated anti-rabbit (Cell Signaling Technology, #7074; 1:2000, Danvers, MA, USA) and anti-mouse (Sigma-Aldrich, A0944, Darmstadt, Germany).

### 4.5. Phylogenetic Analysis

The ClustalW algorithm (BLOSUM62 matrix with a gap opening penalty = 10; gap extension penalty = 0.2) was used in the MEGA X [27] software to calculate all multiple sequence alignments. The neighbor-joining (NJ) method [23] was utilized to construct the phylogenetic trees. 1000 bootstrap replicas [25] were calculated using the Poisson correction method [26]. The phylogenetic trees are drawn in the units of the number of amino acid substitutions per site. The different protein domains were identified with the ExPASy PROSITE online tool (https://prosite.expasy.org/ accessed on 14 December 2020) [24].

## Figures and Tables

**Figure 1 ijms-22-06287-f001:**
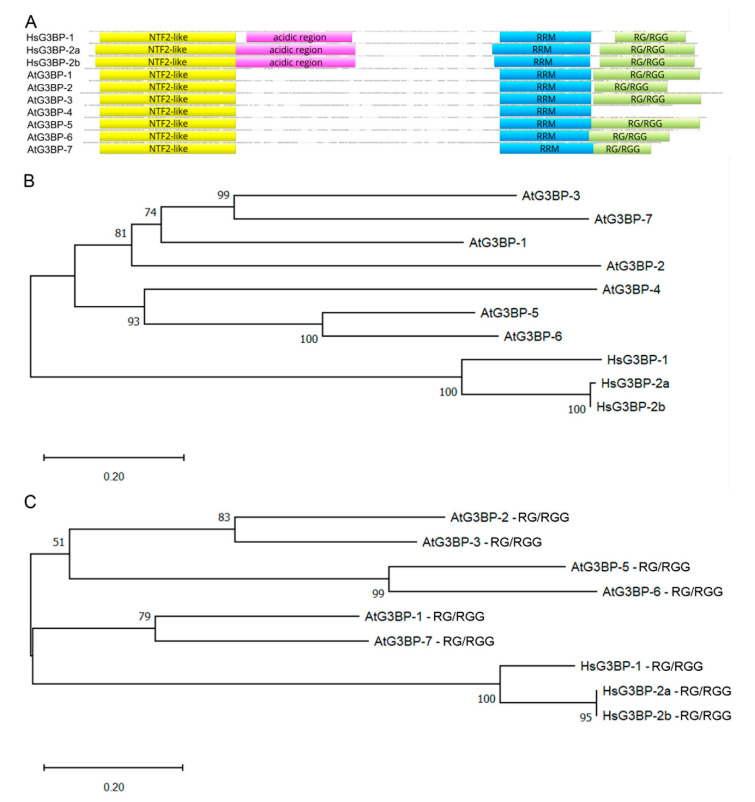
(**A**) Schematic alignment of the HsG3BPs with the G3BPs from *A. thaliana* generated with Geneious Prime^®^ 2020 software. NTF2-like domain, acidic region, RRM and RG/RGG domains are annotated and highlighted with yellow, purple, blue and green boxes, respectively. Domains, regions and motifs were identified using ProSITE [24]. (**B**) Phylogenetic tree of the full-length proteins and (**C**) the RG/RGG domains of the *Arabidopsis* G3BP-family and HsG3BP-1, -2a and -2b. The evolutionary history was inferred using the neighbor-joining method [23]. The optimal tree with the sum of branch length = 4.77 (**B**) and 3.58 (**C**) is shown. The percentage of replicate trees in which the associated taxa clustered together in the bootstrap test (1000 replicates) are shown next to the branches [25]. The trees are drawn to scale, with branch lengths in the same units as those of the evolutionary distances used to infer the phylogenetic tree. The evolutionary distances were computed using the Poisson correction method [26] and are in the units of the number of amino acid substitutions per site. This analysis involved 9 amino acid sequences. All ambiguous positions were removed for each sequence pair (pairwise deletion option). There were a total of 98 positions in the final dataset. Evolutionary analyses were conducted in MEGA X [27].

**Figure 2 ijms-22-06287-f002:**
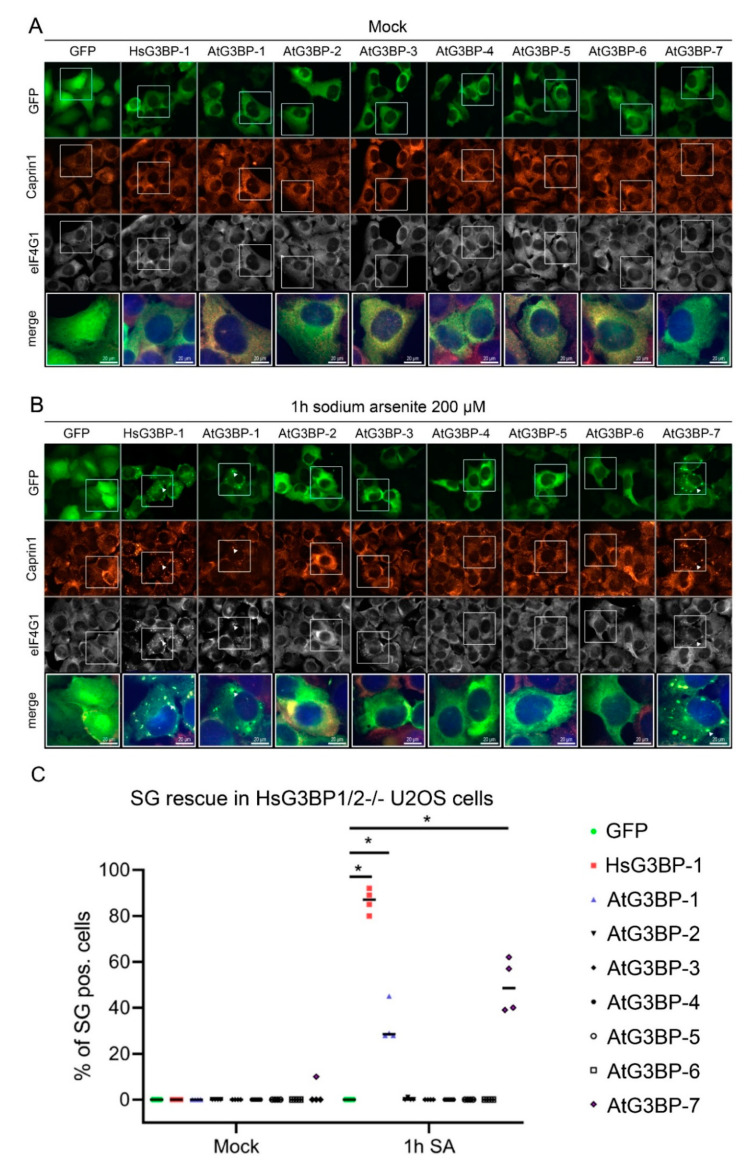
Complementation assay of the AtG3BP family in HsG3BP1/2^−^^/−^ U2OS cells stably expressing the indicated constructs. (A+B) HsG3BP1/2^−/−^ U2OS cells stably transfected with the different EGFP-G3BPs or free GFP under mock conditions (**A**) or after 1 h treatment with 200 µM SA (**B**) stained for Caprin-1 (red), eIF4G1 (grey) and DNA (blue in the merged channel). Exemplary co-localization of G3BP, Caprin-1 and eIF4G1 is marked with an arrowhead. (**C**) Quantitative analysis of SG rescue in cells expressing the indicated construct. The asterisks indicate significance calculated by a Mann–Whitney *U* test (* = *p* < 0.05) compared to the GFP control.

**Figure 3 ijms-22-06287-f003:**
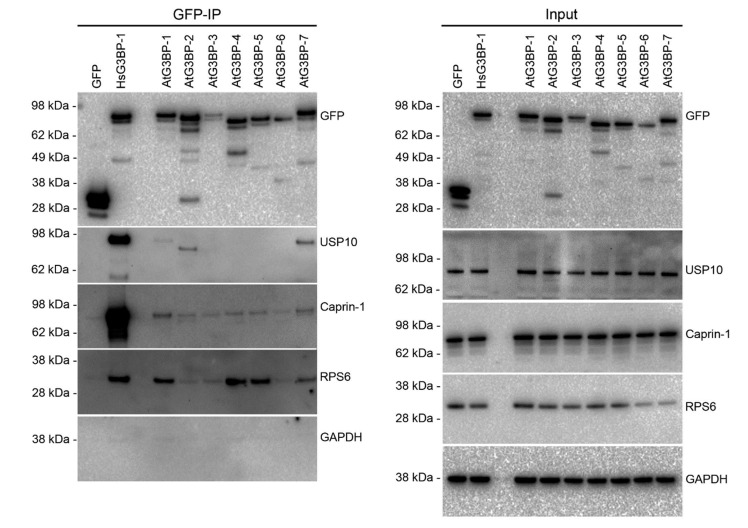
Identification of human interaction partners of the different AtG3BPs. GFP immunoprecipitates and cell lysates were analyzed by Western blot for GFP, USP10, Caprin-1, RPS6 or GAPDH.

**Figure 4 ijms-22-06287-f004:**
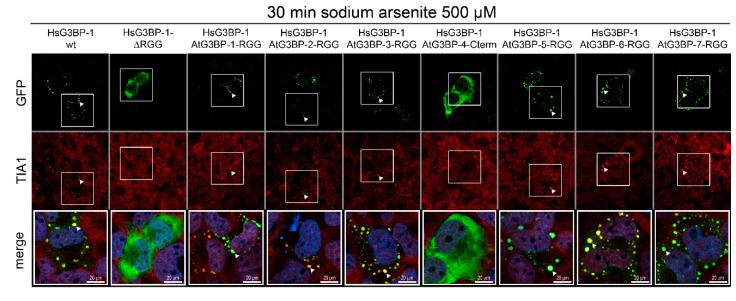
RG/RGG domain exchange. The indicated EGFP::G3BP constructs were expressed in HsG3BP1/2^−/−^ U2OS cells and treated with 500 µM SA for 30 min to induce SG formation. The cells were then fixed and stained for TIA-1 (red) and DNA (blue in the merged channel).

## Data Availability

Not applicable.

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
