# Peer review of "Arabidopsis thaliana* G3BP Ortholog Rescues Mammalian Stress Granule Phenotype across Kingdoms"

_ijms, 2021, doi:10.3390/ijms22126287_

Round 1

Reviewer 1 Report

In this study, AtG3BP protein family were investigated in stress granule through using mammalian cell system. The function of AtG3BP1/7 and RG/RGG domain were identified. All this work is clear and interesting. However, some points need to be addressed:

  1. Compared with HsG3BP, AtG3BPs lack acid region. What is the role of this region, and how to prove it if AtG3BPs have this region?
  2. In this study, the authors used hsg3bp1/2-/- U2OS cells to detect SA induced SG formation of HsG3BP 1 and AtG3BPs. What is the significance of using double mutants? What's the difference between using wild-type animal cells and double mutant in this study?
  3. AtG3BP1 / 7 are similar to HsG3BP 1 in regulating SG formation in animal cells. Do they have similar functions in plant cells? It is suggested to use plant cell system for verification. In addition, the physiological functions of the two in plants can be detected by single mutant or double mutant, if there are plant materials.
  4. The RG/RGG domains of AtG3BP1-7 (except 4) have the same function as HsG3BP1. It is found that RBD domain is the main interaction region between G3BP and eIF4G1 (Yang Xiaodan et al., 2019). How to explain that RG/RGG domains have different interaction mechanisms and functions though they have similar functions? And why AtG3BP2/3/5/6 has not detected the function of forming SG formation?
  5. Why the conditions of fig2B and Fig4 are different? Does the SA concentration and time of treatment affect the SG formation of AtG3BPs? Is there any difference in SG formation activity (time or size?) between different G3BP?
  6. In Fig3 interaction experiment, whether the samples were taken under normal conditions or after SA treatment, the occurrence of SG formation needs SA stress, and the interaction between G3BP and downstream factors may also need SA treatment.
  7. In FigS3, there is a very significant Caprin band in the maker column of B and C, which is very puzzling.

Author Response

Dear reviewer,

we would like to thank you for the helpful comments and your effort to improve our manuscript. Please find attached our answers to all three reviewers. Changes in the manuscript have been indicated within this cover letter.

Sincerely,

Hendrik Reuper, Marc Panas & Hendrik Reuper

Reviewer 2 Report

The manuscript by Reuper et. al. describes the functional analyses of Arabidopsis G3BP proteins in complementing human G3BP knockout cell lines. The authors have found AtGBP1 and 7 are functional and able to form stress granules (SG) within G3BP knockout human cell lines. In addition, both AtG3BPs were shown to associate with many known human G3BP interactors. Moreover, domain-substitution analyses revealed the functional conservation of AtG3BP RGG domains. The work presented here will provide functional insights into the mechanisms that regulate SG formation in Arabidopsis.

I have the following comments concerning the article:

  1. As I understand, both Figures 1 and 4 essentially show the complementation assays for AtG3BPs (Figure 1) or chimeric proteins of human G3BP with RGG from Arabidopsis (Figure 2). If so, I am wondering why different methods were used for both analyses. For example, 1 hour of Sodium Arsenite 200 µM in Figure 1 and 30 minutes 500 µM in Figure 2. Also, Caprin1 and elF4G1 detection in Figure 1 and TIA1 staining in Figure 2. Could you please explain the reasons behind it?
  2. Mock treatment is missing in Figure 4.
  3. Equal loading controls and inputs are missing in the Co-IP experiments (Figure 3 and S3)
  4. From the domain-substitution experiment (Figure 4), the possibility of human RGG alone facilitate SG formation cannot be ruled out. If it can, then the results presented in Figure 4 would not be surprising as many of the AtG3BPs share high homology in RGG with human G3BPs. It would also point to the essential function of domains other than RGG in SG formation in AtG3BPs. Hence it is recommended to include this control or at least mention about this in the discussion section.

Author Response

(The authors gave the same response as above.)

Reviewer 3 Report

This is a very interesting paper shown functional redundancy in the mechanism of stress granules induction in plants and mammals.

Experiments were well design and well done, but some points need to be addressed.

Layout of the figures require modifications.

It will be nice to add some quantitative bioimage analysis to confirmed  observation on figures 2 and 4.   

I would suggest to add proximíty ligation assay to show more precise protein-protein interactions.

Line 85: „stress experiment“? It is better to mention „stress treatments“.

Line 201: „protein mRNA aggregates“ – maybe better is: „proteins-mRNA aggregates“: it is not one protein and it is better to add defice.

Line 220: please, edit: „sevenfold elimination of all G3BPs“.

Line 224: „(data will be presented elsewhere)“ – please, clarify.

Figure 2: please, explain what is blue color; add scale bar for each lines; Maybe move merge after single channels.

Figure 4: please, change layout: it is better to put merge images after single channel. Please, use the same magnification for all images or mention scale bar for each.

Lines 276-283: please, made uniform Sigma-Aldrich links: sometime it is only Sigma-Aldrich, but in some you mentioned St. Louis, MO

Author Response

(The authors gave the same response as above.)

Round 2

Reviewer 1 Report

The authors replied all the comments carefully and provided sufficient evidence for each comment. I suggest it can be accepted.  

Author Response

thanks for your kind comment

Reviewer 2 Report

The authors have satisfactorily addressed my comments concerning the experiments in the manuscript.

Author Response

thanks for your kind comment

Reviewer 3 Report

The authors significantly improve text and answered all points.

Author Response

thanks for your kind comment